# Content Analysis of Talent Policy on Promoting Sustainable Development of Talent: Taking Sichuan Province as an Example

**Huiqin Zhang, Ting Deng \*, Meng Wang and Xudong Chen**

College of Management Science, Chengdu University of Technology, Chengdu 610059, China;
zhanghuiqin@mail.cdut.edu.cn (H.Z.); wang.meng@student.zy.cdut.edu.cn (M.W.);
chenxudong198401@163.com (X.C.)

\* Correspondence: dengting199401@163.com

**Abstract:** Sustainable development of talent refers to the establishment of institutionalized, standardized, and systematic stabilization measures and procedures based on scientific principles. A talent management system is a series of systematic, regular, and systematic practices instead of policy that is short-term, fragmented, and involves special measures and practices. As an effective means of policy analysis, talent policy instruments play a critical role in promoting the sustainable development of talent. This study constructs an analytical framework from the dimensions of policy instruments, policy targets, and policy strength to examine the policy sustainability. It selects 30 talent policies issued by the Sichuan government as the research sample and uses ROST and NVivo software to quantify policy instruments, policy targets, and policy strength as analytical units. The results show that, in the supply-side policy instruments, many "capital investment" policy measures have been used, but the role of "talent information support" in the role of talent development has been neglected. Among the environmental policy instruments, "strategic measures" are used more frequently, and the economic leverage of "tax finance" has not been fully used. Among the demand-side policy instruments, "talent introduction" and "trade control" are used more frequently, but the application of "overseas talent agencies" remains unused. Policy targets focus on talent innovation and talent efficiency but pay less attention to talent flow, and policy strength is in line with strategic adjustments, but policies are not sufficiently sustainable. The formulation of regional talent policy should focus on the operability of policies, improve the structure of policy targets, and lead to sustainable development. In addition, it is necessary to strengthen the implementation of policies and promote the sustainable development of talent.

**Keywords:** talent policy; policy instruments; policy targets; policy strength; sustainable development of talent

## 1. Introduction

To develop high-end manufacturing, high-tech industries, artificial intelligence, international finance, and so forth, each city needs a group of high-end talent with international vision and understanding of international rules and many professionals in finance, trade, culture, and education. In addition, each city needs all types of professional workers and various social service talents. Therefore, China must accelerate the transition from a demographic dividend to talent bonus. However, with the decrease in the birth rate and the imbalance of urban development, some cities in the north and west of China have gradually appeared to be "hollowing". The city's outstanding talents are gradually shifting to developed regions, and the city's core consumption power is gradually shifting to developed regions. Some cities are gradually "hollowing", and the development of the city is followed

by "marginalization". The Chinese economy has experienced rapid development for 40 years and has now reached an inflection point in which the demographic dividend is disappearing and the problem of aging is serious. In 2018, the ratio of people over 65 years old increased to 11.9%.

To develop a city, it is necessary to have consumption, and the overall improvement in consumption power depends on young people. The number of college graduates increased to 8.2 million in 2018—a historical high. The high frequency of "retain and attract talent in the world" appears in the "declaration" of the local government. This talent war is occurring across the country. It presents an increasingly intense trend. The so-called "talent battle" means that since 2017, many cities in China have issued many policies to attract talent. Since 2017, some new first-tier or second-tier cities such as Nanjing, Wuhan, Chengdu, Xi'an, Changsha, Hangzhou, and Zhengzhou have introduced various talent introduction policies. Its broad scope for the introduction of talent and the low thresholds and high levels of treatment are unprecedented. Beginning in the first quarter of 2018, traditional first-tier cities such as Beijing and Shanghai have also introduced corresponding new talent policies. Additionally, more second-tier and third-tier cities have joined the talent war. This "talent battle" has spread to the capital cities of most provinces and regions in China. Although many second-tier and third-tier cities are joining the talent war, first-tier cities in the north—for example, Guangzhou and Shenzhen—also joined. Overall, a critical feature of this talent war is the clearer hierarchy between the requirements of the cities for talent. First-tier cities—for example, Beijing and Shanghai—mainly aim to acquire high-level talent. Second-tier cities generally have a wide range of talent requirements. In addition to high-end talent, they are also relatively loose regarding academic qualifications and professions.

Policy homogeneity is a common problem of talent policy. All local governments are focusing on sending accounts, money, and sending houses to attract talent. A rushing battle of people is often prone to the problem of "thunder and little rain". Additionally, the "talent battle" in some cities in China does not fully consider the natural environment and the humanistic environment for talent growth and talent work. They mainly provide preferential policies from the perspective of treatment and identity. Even if they can attract talent, it is difficult to ensure the good use of talent, and it is more difficult to guarantee the retention of talent. The victory of the "talent battle" is not merely a matter of robbing talent, because cultivating talent is more important. A "talent battle" is not the ability to grab talent but the mechanism to train talent. Selecting, using, educating, and retaining talent all require effective methods and measures. Is a "talent war" sustainable? Can talent be retained and used well? Does each city have such a need? Is there a range of ancillary services? This series of questions has triggered our thinking; that is, it is unsustainable to solve the shortage of talent through policy adjustment. The policy focuses on solving short-term problems, and talent is a hundred-year plan that should be solved through systems and legislation. The preferential policies such as settlement, housing, and financial support are tempting, but in the long run, it is not sufficient to support the sustainable development of talent. For second-tier and third-tier cities, they need to introduce talent and improve institutional mechanisms. Designed according to its own conditions, it is the best talent for the local talent; otherwise, it is a waste of the resources invested and the talent itself, which leads to the unsustainable development of talent.

An effective evaluation of sustainability is critical because it provides a useful framework for improved decision-making on all undertakings, such as policies, plans and programs, and physical undertakings [1]. However, sustainability is not easily measurable because it is not directly indicated as a natural or direct consequence of the reading of the indicators [2]. A widely accepted approach is about developing a framework or model by involving a number of sustainability criteria [3]. This study uses content analysis and statistical measurement to build policy instruments, policy targets, and a policy strength three-dimensional analysis framework, and takes Sichuan Province talent policy as a research sample. This study analyzes the problem and the cause of existing talent policies. Next, this study provides effective advice for the sustainable of policy. Moreover, this study has important theoretical and practical significance for analyzing the political power game in the process of policy

implementation, guiding the formulation of the talent policy system and promoting the sustainability of the development of talent.

## 2. Literature Review

### 2.1. Policy Instruments

Policy instruments are the intermediary between policy targets and outcomes, and this area has become a critical branch of policy research [4]. Talent policy instruments refer to the methods and means adopted by the government to realize the value of talent and enhance the competitiveness of talent. Talent policy instruments are an effective measure and means to achieve the targets of the sustainable of policy [5]. Policy is the process by which the government designs, configures, and uses various policy instruments to achieve targets and must be based on various policy instruments from concept to reality [6].

Scholars have the following classifications of policy instruments. According to the level of intervention, policy instruments are divided into command-and-control policies, market-based incentive policies, and voluntary policies [7]. According to policy goals and means, policy instruments are divided into institutional policies and process policies [8]. The most representative is the classification of policy instruments by Rothwell (1985) [9]. Policy instruments are divided into demand-side policy instruments, supply-side policy instruments, and environmental policy instruments [9].

This study draws on the division of policy instruments by Rothwell [9]. As shown in Figure 1, environmental policy instruments mainly reflect the indirect impact of talent policy on the sustainable development of talent. Supply-side policy instruments are mainly embodied in the sustainable development of talent through talent policy. Demand-side policy instruments are mainly reflected in a talent policy's impetus for sustainable policy. Supply-side policy instruments refer to the government directly providing funds, technology, and other aspects of sustainable policy.

According to the different methods used by the government to promote sustainable policy, supply-side policy instruments are divided into public services, talent infrastructure construction, talent capital investment, and talent cultivation [10], and demand-side policy instruments refer to the government developing and stabilizing the talent market through trade control and service outsourcing. According to the different methods used by the government, demand-side policy instruments are divided into talent introduction, trade control, service outsourcing, and overseas talent institutions [10], and the environmental policy instruments refer to the government influencing the environment for talent development through strategic measures. According to different influencing methods used by the government, environmental policy instruments are divided into talent target planning, talent regulation, financial finance, tax incentives, and strategic measures [10] (Figure 1).

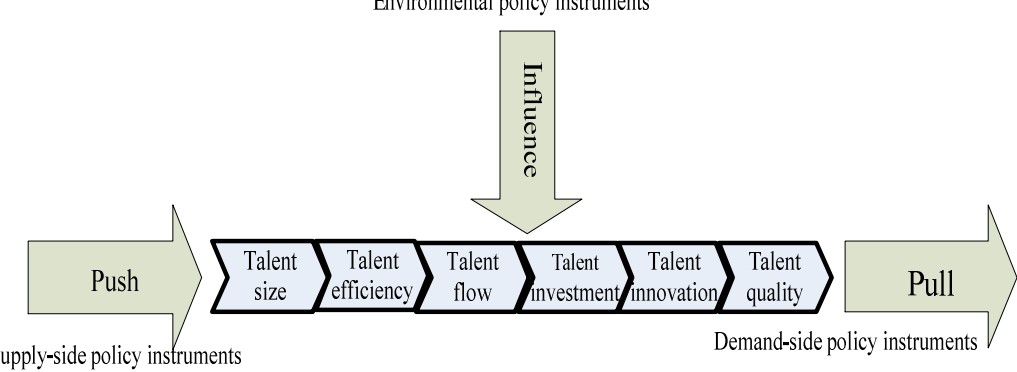

**Figure 1.** Policy instruments for talent development.

### 2.2. Policy Targets

According to a review of the literature, few studies have specifically attempted to investigate sustainable policy in governments at all levels. Most of the studies have focused on performance evaluation analysis of talent policy, but performance evaluation is a comprehensive measurement of talent policy targets. The policy targets are the expected direction of policy implementation, which is the purpose of policy instruments [11]. Different policy instruments can play different roles in the stage of sustainable policy [12]. Therefore, the targets of policy instruments should be considered while analyzing the type of policy instruments. Sun (2014) divided the talent policy evaluation system into four aspects: talent quantity, talent quality, talent input, and talent output [13]. Serban and Andanut (2014) expounded the relevant indicators of the global talent competitiveness index, which are mainly divided into input indicators and output indicators [14]. Zhao and Yu (2017) created a talent policy evaluation system for talent resource competitiveness, talent efficiency competitiveness, and talent environment competitiveness [15]. Zhang and Shi (2017) divided the talent policy targets into five aspects: talent introduction, talent cultivation, talent flow, talent incentive, and talent entrepreneurship [16]. Some scholars (2018) have divided the foreign haze prevention and control talent into talent introduction, talent cultivation, talent use and allocation, and talent evaluation and incentive [17]. This study also draws on the classification in the literature and divides the talent policy targets into talent size, talent efficiency, talent flow, talent investment, talent innovation, and talent quality.

### 2.3. Policy Strength

The policy strength is determined by the administrative level of the policy issuing agency. Usually, the level of policy promulgating units determines the policy strength, and the higher the level of the policy promulgating units is, the stronger the strength of the policy promulgating units is. This phenomenon reflects that the higher the government attaches importance to the sustainable development of talent. The weaker policies are issued by the policy promulgating agencies at lower levels, which also reflects that the lower levels of government attach importance to the sustainable development of talent. At present, the research on policy intensity has mostly drawn on quantitative standards [18,19]. Scholars have made a deeper exploration of policy strength [20,21]. This paper also draws on Peng's thought of quantitative policy and uses a score of one to five for the policy strength (Table 1).

**Table 1.** Quantitative standards for talent policy.

| Scoring Standard | Score |
| --- | --- |
| Sichuan Provincial People's Congress and its Standing Committee enacted local regulations | 5 |
| Sichuan Provincial Government enacted regulations, etc. | 4 |
| Sichuan Provincial Government promulgated the Provisional Regulations, programs, decisions, opinions, methods, standards, etc. | 3 |
| Sichuan Provincial Government under the jurisdiction of commissions, bureaus, offices and other views | 2 |
| Notice, announcement, etc. | 1 |

### 2.4. Construction of the Integrated Framework

Based on the existing talent policy analysis model and framework research, this study increases the policy strength dimension. This paper builds a three-dimensional analysis framework for sustainable policy (Figure 2).

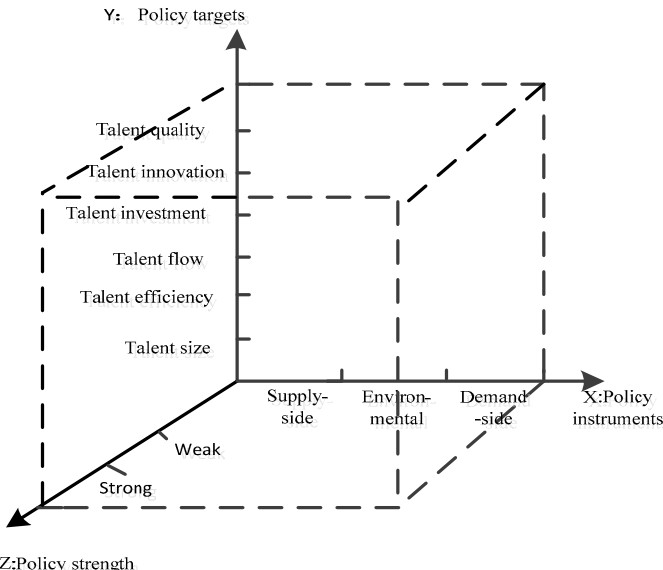

**Figure 2.** Three-dimensional analysis of talent policy.

## 3. Methods

### 3.1. Research Methods

This paper uses a content analysis method that can systematically and objectively quantify as well as qualitatively analyze text content. Content analysis is a scientific, objective, and quantitative analysis method for different sample information content, and its concept was first proposed by Berelson [22]. Content analysis includes six steps: determining research questions, selecting samples, determining analysis units, classifying and encoding data according to research questions, verifying the reliability of encoding, and analyzing data obtained through encoding and drawing conclusions [23]. Content analysis can quantify the qualitative data, use statistical methods to measure the frequency of the analysis analogy and analysis unit, and use figures or charts to express the results of the content analysis. Content analysis is mainly used to reveal the historical changes of policy topics, the choice and combination of policy tools, the main cooperation network of policy process, and other public policy research issues [24]. The content analysis method combines qualitative and quantitative analysis of text content to clearly analyze essential facts and related trends of relevant topics within the policy documents [25], which has been used by many scholars to analyze policy. For example, Farchi and Salge (2017) used the content analysis method to trace the evolution of the innovation concept in policy discourse based on the analysis of 21 key policy documents published or commissioned by the English Department of Health between 1948 and 2015 [26]. Liao (2018) used the content analysis method to analyze Chinese environmental policies for promoting enterprise environmental innovation [27]. Heslinga et al. (2018) applied content analysis to local documents from 1945 to 2015 to identify fluctuations and shifts in the focus of these documents. This content analysis was augmented with semi-structured interviews with local experts and other key stakeholders [28]. Ledoux et al. (2018) used content analysis to review the policy framework for migrants' access to healthcare in Spain, Portugal, and Ireland, countries with a long history of immigration, to identify lessons to be learned for policies on migrants' health [29]. Laestadius et al. (2019) examined e-liquid marketing on the visual social media platform Instagram, on which users have created significant amounts of e-cigarette-related content with the content analysis [30]. This paper also used content analysis to analyze whether talent policy can promote the sustainable development of talent.

This study mainly uses NVivo11 software to encode and analyze text. There are two common coding methods. The first method is to set the coding nodes according to the research theme and then form a research framework and a more detailed coding through the deep mining of nodes.

The second method is according to the steps of open encoding, spindle encoding, and selective encoding, namely, the text is first encoded meticulously and then grouped into several sub-nodes to form related categories [31]. This paper mainly adopts the first coding method. Our specific operation process is as follows: (1) we determine analysis dimensions of talent policy analysis; (2) we use ROST NAT4.6 to analyze the social network and the semantic network of policy text; and (3) based on the established analysis dimensions, we use NVivo 11 to establish nodes, code them, and then quantify the results of the coding analysis and draw conclusions.

## 3.2. Data Collection

The data used in this paper are policy documents from sources such as the Sichuan Talent Government, the Sichuan Talent Work Network, and the Sichuan Political Consultative Conference Network. We select the policy documents according to the following principles: (1) the issuing unit was the Sichuan provincial government and its related departments; (2) the policy selected documents were directly related to the reform of institutional mechanisms for talent development policy; (3) the policy documents were released from 2013–2017; and (4) with the gradual development of the concept of sustainable development and with emphasis on talent, the number of talent policies has increased, as has the number of departments issuing such policies. A number of departments had jointly issued policy texts, and 30 valid policy documents were obtained (Figure 3). Through the networked visual analysis of Sichuan Province's talent policy from 2013–2017, we found that "talent" has the highest frequency and is most closely related to other words. It can also prove that the policy text selected in this article is closely related to talent.

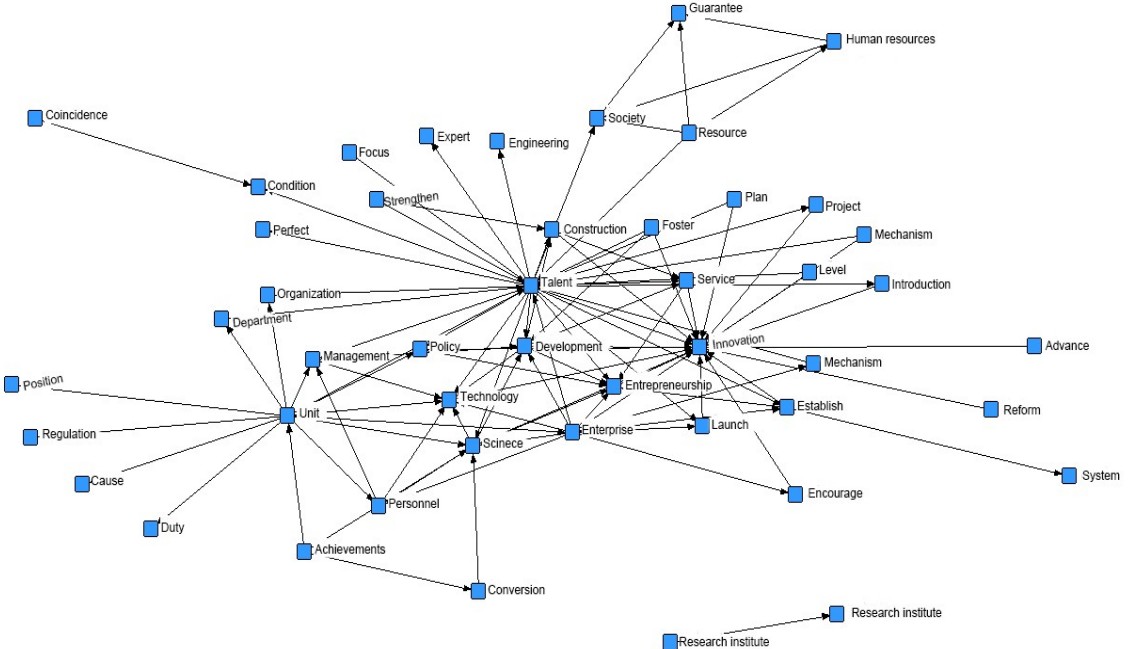

**Figure 3.** Talent policy words network association view.

## 3.3. Policy Texts Encoding

The word network association view obtained by ROST analysis shows a macroscopic view of the talent policy of Sichuan Province from 2013–2017 as a whole, but it is impossible to analyze the text content in detail. Therefore, this article continues to use NVivo software to further explore the text to ensure the integrity of the research results. To maintain the integrity of the data and prevent the omission of the main information, this paper adopts the line-by-line coding method for the research text. All the policy texts are approximately 140,650 words, and we divide the policy text into the smallest unit with each rule.

After determining the research dimensions and keywords, NVivo11 software is used to build nodes and encode text. First, "supply-side policy instruments", "environmental policy instruments", "demand-side policy instruments", "policy target", and "policy strength" are defined as tree nodes; second, sub-nodes are established under tree nodes; third, the words or sentences associated with the nodes are placed under the corresponding nodes; and fourth, the coding level of the "reference point-node-tree node" is formed (Table 2). In NVivo, the commonly used percentage agreement and k-factor are encoded as a method of reliability analysis. By inviting another coder to extract one text from the three policy texts, and then by re-encoding it using the code consistency check function in NVivo, the code is found to have high credibility.

**Table 2.** Encoding process example table.

| Tree Node | Node | Reference Point |
|---|---|---|
| Supply-side policy instruments | Talent cultivation | We establish an entrepreneurial training base, regularly organize training objects to carry out capacity training, and enhance their strategic thinking and modern management ability. |
| Demand-side policy instruments | Talent introduction | We encourage the overall relocation of the headquarters of foreign businesses to return to Sichuan or the establishment of new regional headquarters, functional institutions, and home development project headquarters bases in Sichuan. |
| Environmental policy instruments | Financial finance | For high-level talent and teams who innovate and start businesses from outside the country or province, and who fulfill the requirements stipulated in the Special Support Measures for high-level talent in Sichuan Province, priority should be given to the provincial "Thousands of Persons Plan", and a one-time home subsidy of 2 million yuan for individuals and a project subsidy of 5 million yuan for teams should be given at the highest level. |
| Policy targets | Talent flow | We encourage migrant workers and entrepreneurs who successfully start their own businesses to fully tap into and use their comparative advantages in terms of resources and factors in their hometowns to transfer suitable industries to their hometowns for re-entrepreneurship and re-development. |

## 4. Results

### 4.1. Reference Point Number and Coding Density Statistics

Encoding the Sichuan talent policy text in NVivo, we obtain the reference points of each node (Table 3). On the premise that the total amount of text is consistent, the more coding reference points there are, the more adequate the disclosure of the information noted is. The reference point number is a reflection of the absolute number of nodes distributed in the text, and the coding density can represent the relative proportion of the node distribution. Using the NVivo coded query function, we determine the source of the node's encoding and its encoding density. Table 4 summarizes the coding density of policy texts at different nodes in different periods. The higher the coding density of a node is, the greater the proportion of the node in the text is. Notably, this information is important for policy makers.

**Table 3.** Reference point statistics.

| Tree Node | Node | Number |
|---|---|---|
| Supply-side policy instruments | Talent cultivation | 30 |
| | Talent information support | 7 |
| | Talent infrastructure construction | 19 |
| | Capital investment | 25 |
| | Public service | 11 |
| Environmental policy instruments | Talent target planning | 65 |
| | Regulatory control | 46 |
| | Finance | 15 |
| | Tax incentives | 4 |
| | Strategic measures | 79 |
| Demand-side policy instruments | Talent introduction | 8 |
| | Service outsourcing | 2 |
| | Trade control | 3 |
| | Overseas talent institutions | 0 |
| Policy targets | Talent size | 53 |
| | Talent efficiency | 60 |
| | Talent flow | 37 |
| | Talent investment | 43 |
| | Talent innovation | 70 |
| | Talent quality | 51 |

**Table 4.** Code coverage statistics.

| Tree Node | Node | Coverage |
|---|---|---|
| Supply-side policy instruments | Talent cultivation | 0.10 |
| | Talent information support | 0.02 |
| | Talent infrastructure construction | 0.06 |
| | Capital investment | 0.08 |
| | Public service | 0.04 |
| Environmental policy instruments | Talent target planning | 0.21 |
| | Regulatory control | 0.15 |
| | Finance | 0.05 |
| | Tax incentives | 0.01 |
| | Strategic measures | 0.25 |
| Demand-side policy instruments | Talent introduction | 0.02 |
| | Service outsourcing | 0.01 |
| | Trade control | 0.01 |
| | Overseas talent institutions | 0 |
| Policy targets | Talent size | 0.17 |
| | Talent efficiency | 0.19 |
| | Talent flow | 0.12 |
| | Talent investment | 0.14 |
| | Talent innovation | 0.22 |
| | Talent quality | 0.16 |

*4.2. Policy Instruments*

The categories of policy instruments based on the analysis are shown in Table 3.

The most used policy instruments to promote the sustainable development of policy implementation are environmental policy instruments (209 items) and supply-side policy instruments (92 items), and the least used are demand-side policy instruments (13 items). The government's dependence on various policy instruments is different in promoting sustainable development of talent. The government is inclined to continually optimize the external environment for talent development

through indirect methods. The low frequency of demand-side policy instruments will lead to the absence of a stable market environment for the sustainable development of talent.

### 4.2.1. Overuse of Environmental Policy Instruments for Sustainable of Policy

The frequent use of environmental policy instruments fully reflects Sichuan's emphasis on sustainable policy and is intended to lead to a sustainable environment for talent development through macro-policy; however, some talent policies are ambiguous and have insufficient maneuverability. Specifically, in environmental policy instruments, strategic measures, talent target planning, and regulatory control are indistinguishable and account for 37.9%, 31.1%, and 22.0%, respectively, of environmental policy instruments. Finance is 7.1%, and tax incentives are only 1.9%.

There is a relative shortage of tax policy instruments. Finance, as the basic guarantee for the sustainable development of talent work, has not received due attention. In terms of regulatory control, although there are many measures, a vacuum zone remains, such as how to ensure a fair talent-sustainable development environment and carry out the follow-up work after the introduction of high-level talent. With the advancement of society, the quality of both sexes is constantly improving, and the development guarantee of female talent should also be a critical aspect of talent policy planning. Moreover, the increasingly prominent role of talent has made the competition among high-level talent intensify. Therefore, how to create a satisfactory policy environment for the sustainable development of talent while performing a satisfactory job of talent introduction is also a focus of follow-up talent work.

### 4.2.2. Insufficient Use of Supply-Side Policy Instruments for Sustainable of Policy

The policy of talent cultivation in supply-side policy instruments shows a greater advantage. One reason is that in the era of the knowledge-based economy, higher requirements are put forward for the cultivation of talent. It is necessary to cultivate their higher learning and innovation abilities to make them achieve sustainable development. Talent cultivation in the new era is imperative, and different types of talent require different training methods. The introduction of talent cultivation policy reflects that the government follows the growth of talent and attaches importance to the sustainable development of talent. A very direct and effective means to promote the sustainable development of talent is by cultivating talent in different fields in an appropriate manner. From the proportion of policy instruments used, talent cultivation investment accounts for the highest proportion. It fulfills the government's needs for cultivation of different types of talent. The development of talented public services and talent development infrastructure is relatively low, and the availability of public services is relatively small. These factors show that the policies of talent infrastructure construction and public service system construction in Sichuan Province are still insufficient, and the public services available in the process of the sustainable development of talent are not perfect. In the information age, the support and the availability of talent information is the key to the sustainable development of talent. The related work in this area requires further deepening and strengthening.

### 4.2.3. Less Use of Demand-Side Policy Instruments for Sustainable of Policy

Demand-side policy instruments have ensured sustainability in a certain manner and broadened the source of talent. In the sustainable development of talent, demand-side policy instruments are more direct and faster than environmental policy instruments; they improve the quality of talent and ensure the sustainable development of talent. However, demand-side policy instruments have the lowest proportion, and the specificity and the clarity of the policy content are weak. The inadequate application of demand-side policy instruments reduces their instrumental value in creating stable markets for the sustainable development of talent. In demand-side policy instruments, talent introduction has the highest proportion, which reflects the government's emphasis on the sustainable development of talent. Instruments such as service outsourcing, trade control, and overseas talent institutions are scarce. In the new era, the establishment of overseas talent institutions is critical to the sustainable

development of talent. However, these policy instruments are clear and the most direct means to promote the sustainable development of talent.

### 4.3. Policy Targets

On the basis of the dimension analysis of policy instruments, we add the policy targets' dimension measurement indicators of talent policy and study the dimensions of talent size, talent efficiency, talent flow, talent investment, talent innovation, and talent quality. This study analyzes the role of talent policy targets in promoting sustainable development of talent and searches for policy instruments that must be strengthened in talent policy targets. After NVivo coding, through the content analysis of each coding point, the statistical results of the talent policy targets and policy instruments are formed in two-dimensional distribution (Table 5).

There are 53 policy rules for achieving the target of talent scale, accounting for 16.9% of the overall policy rules; 60 policy rules for achieving the target of talent efficiency, accounting for 19.1% of the overall policy rules; and 37 policy rules for achieving the target of talent flow, accounting for 11.8% of the overall policy rules. There are 43 policy rules for achieving the targets of talent investment, accounting for 13.7% of the overall policy; 70 policy rules for achieving the target of talent innovation, accounting for 22.3% of the overall policy rules; and 51 policy rules for achieving the target of talent quality, accounting for 16.2% of the overall policy rules. Sichuan Province promotes the sustainable development of talent mainly through talent innovation and the improvement of talent size, which is closely related to the competition of talent in various provinces and cities. The impetus of innovation is the main line of the sustainable development of talent. By strengthening talent innovation and expanding the talent size, talent will become the foundation of reform and innovation, enhancing its sustainability. Talent efficiency and talent flow account for almost the same proportion of policy targets. These two policies are mainly reflected in the evaluation of talent value through talent efficiency and ultimately promote the sustainable development of talent. The government's use of policy instruments increased the talent scale and introduced policies on talent quality. However, the government should introduce overseas talent and encourage the flow of talent at the grassroots, inter-regional, inter-industry, or inter-firm level. Generally, in promoting the sustainable development of talent, the policy targets mainly use environmental policy instruments.

**Table 5.** Code coverage statistics.

| | Policy Targets | Talent Size | Talent Efficiency | Talent Flow | Talent Investment | Talent Innovation | Talent Quality |
|---|---|---|---|---|---|---|---|
| **Policy Tool** | Supply-side policy instruments | 20 | 13 | 20 | 15 | 14 | 10 |
| | Environmental policy instruments | 27 | 47 | 14 | 27 | 55 | 39 |
| | Demand-side policy instruments | 6 | 0 | 3 | 1 | 1 | 2 |

### 4.4. Policy Strength

In Figure 4, the average strength of the policies issued by Sichuan Province since 2013 has shown an increasing trend, which reflects the importance of the sustainable development of talent, and also shows that the Sichuan government has increased its awareness of the importance of the sustainable development of talent. From the policy strength of talent policy promulgation in Sichuan Province, 2016 is a peak. This result is because in March 2016, the Central Committee of Communist Party of China issued the "Opinions on Deepening the Reform of the System and Mechanism of Talent Development". The provincial talent policies were successively proposed and implemented, which further contributed to the unprecedented growth rate of the policy in 2016. The adjustment of talent policy has become a crucial source for the sustainable development of talent in Sichuan Province. However, the policy has experienced ups and downs, which reflects the absence a scientific and systematic understanding of the sustainable development of talent. The understanding of the law to promote the sustainable

development of talent is not sufficiently deep. Thus, the sustainability and the coherence of policies must be strengthened.

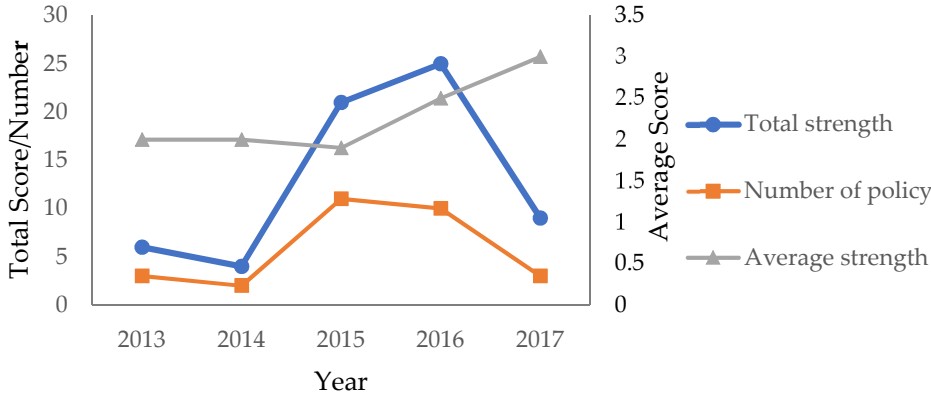

**Figure 4.** Historical evolution of policy.

*4.5. Cross-Analysis of Policy Instruments, Policy Targets, and Policy Strength*

In Figure 5, although sustainable policy is mainly based on the use of environmental policy instruments, the policy strength is significantly lower than the supply-side policy instruments. Environmental policy instruments and supply-side policy instruments are mainly used in talent scale, talent efficiency, and talent innovation. The average policy intensity on these three targets is higher than other policy targets, and the policy of talent mobility is the lowest. Therefore, we observe an inconsistency between policy instruments and policy targets in the current policy system of promoting the sustainable development of talent in Sichuan Province, mainly there is because no long-term guidance in the policy system.

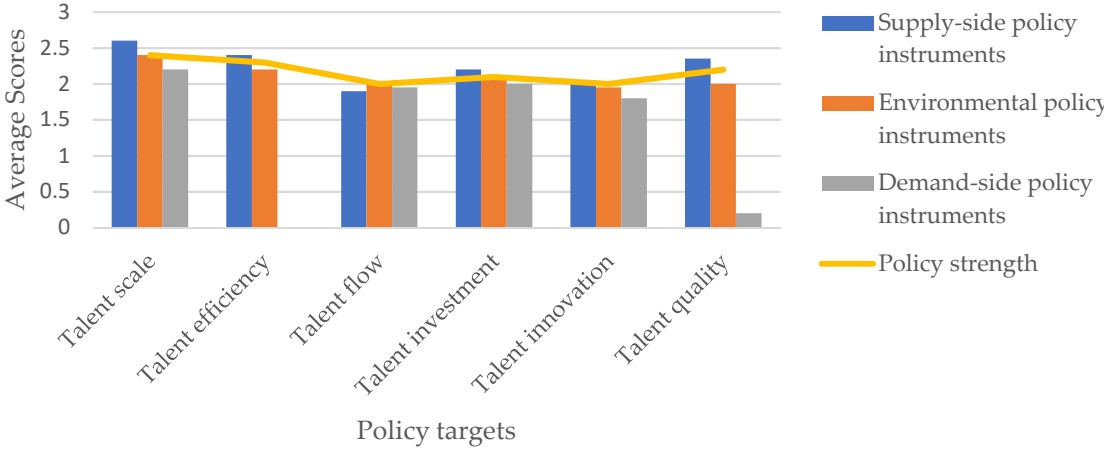

**Figure 5.** Three-dimensional analysis of talent policy.

In the process of using policy instruments to achieve individual talent targets, the government mainly uses environmental policy instruments to promote talent innovation, optimize the talent environment, and improve the quality of talent. These crucial means allow the government to promote the sustainable development of talent. However, excessive dependence on policy instruments to achieve a certain policy target may not result in the desired effect. To create talent quality and talent efficiency policy, the government has used supply-side policy instruments to form a "policy crowding" phenomenon to a certain extent, and this will be accompanied by a series of exogenous problems caused by these policy fights and conflicts. The three-dimensional statistics on the reform policy of the talent development system in Sichuan Province shows the crowdedness of the policy system and reflects the gaps and deficiencies of the whole system. The use of demand-side policy instruments is

minimal in the process of realizing the goal of the sustainable development of talent. The demand-side talent policy instruments are more direct and faster in the effort to achieve the sustainable development of policy. Expand the talent scale, increase the efficiency of talent, and promote the flow of talent by mainly relying on the synergy of supply-side, demand-side, and environmental policy instruments.

## 5. Discussion

### 5.1. Policy Instruments and Sustainable of Policy

Our study shows that in the supply-side policy instruments, many "capital investment" policy measures have been used, but the role of "talent information support" in the role of talent development has been neglected. Among the environmental policy instruments, "strategic measures" are used more frequently, and the economic leverage of "tax finance" has not been fully used. Among the demand-side policy instruments, "talent introduction" and "trade control" are used more frequently, but the application of "overseas talent agencies" remains blank.

In the information age, the acquisition and effective use of talent information is the basis and guarantee of doing a good job in talent work. All types of talent databases and information databases are the support, and as such, they support the effective docking of supply and demand information of scientific and technologically innovative talent and can also provide more complete services for the exchange and the development of talent. To guarantee comprehensive information and perfect service, we should integrate the information of talent, projects, policies, funds, and other factors in the content construction of information platforms, use the systematic and interactive role of information platforms, and expand the amount of information. We should also provide channels for information exchange and interaction between the supply and demand sides. In terms of the type of information platform construction, it should include the information base serving the orderly flow of talent and the scientific allocation of talent as well as establish an integrated platform to promote the flow of information among industries, scientific research, and innovation work and to facilitate the use and transformation of scientific and technological achievements. We also found that the imperfect and incomplete policy content may lead to the imperfection of talent information support. There are no specific requirements for the quality of information release and the perfection of platform construction in the policy items, nor are there any regulations or explanations for the comprehensiveness of information release content and the interactive role of platforms. The problem is that the content of information platform issuance is not comprehensive and perfect, and the quality of platform construction must be further improved.

In Table 3, we observe that the main purpose of environmental policy instruments is to provide a satisfactory policy environment for the sustainable development of talent and to open channels for the sustainable development of talent through indirect means. Compared with other policy instruments, the effectiveness of this policy instrument is slower, and indirectness is its main feature. Environmental policy instruments are indispensable, but there has been overuse of these instruments in Sichuan Province. Moreover, in environmental policy tools, policy tools are highlighted as strategic measures; on the one hand, this is caused by the short-term behavior of the government, and on the other hand, it is caused by the imperfection of the system. The government attaches great importance to the construction of a talent management system and mechanism. It also formulates comprehensive policies to guide the use and selection of talent, improves the degree of legalization, and effectively standardizes and optimizes the sustainable development of talent. However, "tax incentives" have not been a focus at the level of policy-making and have been applied less often, and the policy ambiguity is strong. As a critical guarantee regarding the sustainable development of talent work, the absence of "tax finance" will significantly reduce the effectiveness of environmental policy instruments. Therefore, policies in this field must be proposed and perfected.

To some extent, demand-side policy instruments are more direct, effective, and efficient than supply-side and environmental policy instruments in promoting the sustainable development of talent. From service outsourcing to trade control, the direction is clear, that is, it is the most direct

means to promoting the development of talent undertakings and ensuring the sustainability of talent development. The establishment and management of overseas talent institutions have broadened sources and channels of talent, guaranteed quality of talent, and promoted exchange and flow of talent. However, from the analysis results, the use of demand-side policy instruments is generally low, and the use of trade control, service outsourcing, overseas talent institutions, and other instruments is particularly inadequate. At present, Sichuan Province's talent management work is mostly in the stage of self-service and does not have the ability or the experience to train and use social forces to engage in talent service work, and the policy guidance, in this respect, is not strong. Most of the policy documents that have been issued have only programmatic provisions, advocating cooperation between the government, enterprises, and social institutions to improve the quality and the efficiency of personnel services and reduce the cost of social management through various means, such as outsourcing and crowdsourcing. However, the specific means and methods of service outsourcing are not specified in detail, which leads to the inoperability of follow-up policies. The use of demand-side policy instruments will make industrial development and markets more closely integrated and embedded to create a more favorable industrial and social environment for the development of talent research work and play a role in promoting the sustainable development of talent. By contrast, the absence of demand-side policy instruments and general content weakens the overall guiding role of provincial policies; thus, the follow-up improvement of demand-side policy instruments should be the focus of government policy-making efforts.

### 5.2. Policy Targets and Sustainable of Policy

The construction of an innovative talent team is of great importance. The macro level affects national economic development and international competitiveness, and the micro level relates to the development of various organizations. Therefore, the management of scientific and technologically innovative talent is a global endeavor that requires strategic vision and overall planning. To determine the policy targets is to forecast the total supply and demand of scientific and technologically innovative talent, to analyze the supply and demand of various professional fields, to determine the source of scientific and technologically innovative talent, and to put forward the corresponding attracting measures, considering the needs of current and long-term development of the region. The scientific clarity of targets is related to the implementation of future policies and the sustainable development of talent.

The study finds that the overall planning and assumptions for the flow of talent are less mentioned in the policy. It is also determined by the nature of the policy targets that generally belong to the strategic policy category. However, among the only policy targets, there are more ideas and plans for talent innovation and improvement of talent efficiency but fewer plans for talent flow and investment as well as imperfect planning system and content. Policy targets are scattered in various policies. First, they weaken the importance of policy targets and are not conducive to the follow-up implementation of policies. Second, the policy is not systematic and has a certain impact on the scientific nature of the goal of talent-team building. In addition, problems remain in the policy, such as unclear goal setting, vague content, and difficulties with measurement.

### 5.3. Policy Strength and Sustainable of Policy

We also observed that the policy strength is in line with strategic adjustments, but policies are not sufficiently sustainable. The increase in policy intensity is significantly higher than the number of policies, which indicates that in these peak years, Sichuan Province introduced more talent policies and issued higher levels of policies. Additionally, most of the personnel policies with a strong legal effect have been published and implemented at an early stage. Weak legalization of policy is unfavorable to the stability and continuity of talent policy and inevitably leads to the convergence or conflict between talent policies because of the absence of guidance and norms of programmatic policies. Policy intensity that is too low is detrimental to the systematic distribution of talent policy. In addition, in 2015 and 2016,

although more talent policies were issued, most were the opinions of the commissions, the bureaus, and the offices under the jurisdiction of the Sichuan Provincial Government, which led to the low average policy strength. This analysis shows that the change in the annual average effectiveness of talent policy in Sichuan Province is less affected by the legal effect of the policy. The poor policy strength will hinder the strategic and the integral design of the talent policy system and cannot produce more effective impetus for the development of talent.

*5.4. Limitations and Further Research Directions*

This research has made progress, but there are limitations, and further research is still required. First, this study categorizes the three types of policy instruments according to various measures, but the categorizations are not very detailed. Second, the identification process of policy target, policy instrument, and policy strength patterns is cumbersome. Some steps require the support of expert knowledge.

We anticipate further studies in at least two research directions. First, studies should focus on the construction of knowledge maps for specific policy fields to describe the relationship between policy subjects, policy objects, policy targets, and policy instruments to help deepen policy analysis. The second step to focus on is the development of policy text oriented, machine-based named entity identification methods to automate the identification of policy targets, policy instruments, and policy strength to reduce the need for deep knowledge of the policy field.

## 6. Conclusions and Policy Recommendations

This study constructs an analytical framework from the dimensions of policy instruments, policy targets, and policy strength to examine policy sustainability. The results show that in the supply-side policy instruments, many "capital investment" policy measures have been used, but the role of "talent information support" in the role of talent development has been neglected. Among the environmental policy instruments, "strategic measures" are used more frequently, and the economic leverage of "tax finance" has not been fully used. Among the demand-side policy instruments, "talent introduction" and "trade control" are used more frequently, but the application of "overseas talent agencies" has not occurred. Because the use of single policy instruments may have incomplete functions, the three policy instruments of environment, supply, and demand should be coordinated [32]. It should focus on the use of the types of supply and demand policy instruments, and environmental policy instruments play more of a supporting role [32,33]. Policy targets focus on talent innovation and talent efficiency but pay less attention to talent flow. Policy strength is in line with strategic adjustments, but policies are not sufficiently sustainable.

Environmental policy instruments account for the vast majority of Sichuan's talent policy instruments. In response to this situation, government departments should strengthen the detailed requirements of relevant regulations and introduce support for talent policies. Most of the existing policies and measures, such as talent target planning and strategic measures, provide guidance for the development of talent undertakings at the macro level or create a large environment, and the ambiguity and uncertainty of policy content greatly reduce the practical feasibility of policy practice. In addition, the use of tax incentives and financial policy measures is almost absent, and the role of economic leverage has not been fully used by the government. Specifically, on the one hand, it is necessary to clarify the detailed requirements of relevant provisions, such as strategic measures, talent target planning, and regulation control, and to strengthen the relevant policies; on the other hand, tax and financial policy measures in environmental policy instruments should be taken seriously, and through the adjustment of financial and tax policies, the role of economic leverage and market mechanism should be vigorously used to promote it. The rational and effective development of human resources is a critical guarantee for the sustainable and healthy development of human resources.

Optimize the use of supply-side policy instruments and strengthen the supportive role of health care factors. The government should pay attention to the supporting role of public services,

infrastructure construction, and other health factors for the sustainable development of policy. In terms of talent cultivation, it is necessary to fully mobilize the enthusiasm and the participation of enterprises and companies, and tap and use their potential and advantages in the field of talent cultivation. It should further innovate the training mode and promote it to become one of the main subjects of talent cultivation. In terms of infrastructure construction, based on the background of knowledge-based economy globalization, the government should strengthen the exchange and support of talent information to guide and promote talent interaction and development and further support and strengthen the construction and improvement of talent-related infrastructure to provide a critical platform guarantee for the development of talent undertakings. In terms of public services, the government should further strengthen services and reduce the ambiguity and uncertainty of policy formulation.

Promote the use of demand-side policy instruments and play a direct role in sustainable policy. Compared with supply-side policy instruments and environmental policy instruments, demand-side policy instruments have a more direct role in the sustainable development of talent. Demand-side policy instruments can expand the talent market and drive the development of talent. The use of talent introduction policy instruments is relatively high, and the use of service outsourcing and trade control policy instruments is relatively small, which reduces the government's support for enterprises or scientific research institutions to a certain extent. The government should strengthen the talent service outsourcing policy and formulate and improve the talent market system. The government should also attach importance to the importance of overseas talent institutions in the development of talent undertakings. They can encourage scientific research institutes and enterprises to establish talent cooperation bases overseas and introduce excellent foreign resources.

Improve the policy target structure and lead sustainable development of policy. On the one hand, the expansion of the talent scale and the improvement of talent quality will be promoted simultaneously. At present, the scale of talent in Sichuan Province has been greatly improved. However, a situation where people do not make full use of the talent occurs from time to time, which deviates from the original intention of training talent and leads to the waste of resources. Therefore, the government should pay attention to the cultivation of talent quality on the basis of expanding the scale of talent, and both should be promoted together to shape more top-notch talent and enhance the overall competitiveness of the talent team. On the other hand, the government should encourage people to move and influence people to innovate. Innovation is the subject of this unavoidable era of rapid development. Innovation requires the flow and integration of knowledge. People who are knowledge carriers must carry the flow. Sustainable mobility is a key concept in the literature [34–36]. The government policies should focus on the topics of urban development and talent flow [37]. Moreover, the government should encourage the flow of talent in different positions, different organizations, different regions, and different countries to provide material and institutional guarantees—increase the incentive mechanism for talent innovation and flow of the enthusiasm for talent innovation, fully use the effectiveness of human resources, and continually improve the efficiency of talent.

Strengthen the implementation of policies and promote the systematization of talent policies. In the process of formulating and implementing the follow-up talent policy, the government should appropriately strengthen the policy enactment and improve the legal level of the policy. The government should organize various talent policies and promulgate, formulate, and implement a comprehensive talent policy. In all aspects and stages of the development of the talent cause, it is necessary to issue specific relevant policy measures and implement them. It will take some time for the formulation and implementation of talent policy to show its effect. Therefore, when formulating talent policy, the government should understand the trends of talent development, long-term strategies, and their effects after implementation. We should regularize and institutionalize the design, the formulation, and the implementation of personnel policies. We should absorb all types of talent blindly and focus on the long-term and systematic requirements of talent development. Additionally, it is necessary to sort out the existing talent policies, enhance the guidance of the policy system, and understand the

coordination between different policy instruments and policies. Moreover, the government should increase the effect achieved by the promulgation of policies and promote the implementation of the talent policy.

**Author Contributions:** H.Z. established the framework of the literature analysis system; T.D. was responsible for the overall writing process; M.W. used Nvivo to quantitatively analyze talent policy; X.C. reviewed and approved the final version.

**Funding:** This research was funded by the National Natural Science Foundation of China, grant number 71571149; by the Philosophy and Social Science Planning Program of Chengdu, grant number 2018A05; by the Social Science Research Program of Sichuan, grant number SC18B006; by the Soft Science Research Program of Chengdu, grant number 2017-RK00-00349-ZF; by the Soft Science Research Program of Sichuan, grant number 2018ZR0052; and by the Government Research Program of Sichuan in 2019.

**Acknowledgments:** We sincerely thank the anonymous reviewers for their useful comments and suggestions.

**Conflicts of Interest:** The authors declare no conflict of interest.

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
