# Peer review of "Content Analysis of Talent Policy on Promoting Sustainable Development of Talent: Taking Sichuan Province as an Example"

_sustainability, doi:10.3390/su11092508_

Round 1
Reviewer 1 Report
I confirm my previous evaluation. This work does not aim the final scope of sustainability. Please you can conduct a simple research within the manuscript. Its presence in abstract and conclusion is extremely more high than rest of the paper. It indicatrs clearly that the "sust" is not the core of this work.
Author Response
[Comment 1]: This work does not aim the final scope of sustainability. Please you can conduct a simple research within the manuscript. Its presence in abstract and conclusion is extremely more high than rest of the paper. It indicatrs clearly that the "sust" is not the core of this work.
[Response]: Thank you for your comments. The theme of this paper is to study the sustainability of policy to promote the sustainability development of talent. This study constructed an analytical framework from the dimensions of policy instruments, policy targets, and policy strengths to examine the policy sustainability. Then, we selected the talent policies issued by the Sichuan government as the research sample and use ROST and NVivo software to quantify the policy sustainability. To highlight the theme of the study, we have added a description of sustainability. We will respond to your comment point by point.
(1) We have added a description of sustainability and some relevant literature in the introduction. The following revisions were made.
1. Introduction
This series of questions has triggered our thinking, that is, it is unsustainable to solve the shortage of talent through policy adjustment. The policy focuses on solving short-term problems, and talent is a hundred-year plan, which should be solved through systems and legislation. The preferential policies such as settlement, housing, and financial support are tempting, but in the long run, it is not sufficient to support the sustainable development of talent. For second-tier and third-tier cities, they need to introduce talent and improve institutional mechanisms. Designed according to its own conditions, it is the best talent for the local talent; otherwise, it is a waste of the resources invested and the talent itself, which leads to the unsustainable development of talent.
An effective evaluation of sustainability is critical because it provides a useful framework for improved decision-making on all undertakings, such as policies, plans and programs, and physical undertakings [1]. However, sustainability is not easily measurable because it is not directly indicated as a natural or direct consequence of the reading of the indicators [2]. A widely accepted approach is about developing a framework or model by involving a number of sustainability criteria [3].
[1]Gibson, R.B.; Robert, B. Beyond the pillars: sustainability assessment as a framework for effective integration of social, economic and ecological considerations in significant decision-making. Journal of Environmental Assessment Policy and Management. 2006, 8(3), 259-280.
[2]Gagliardi, F.; Roscia, M.; Lazaroiu, G. Evaluation of sustainability of a city through fuzzy logic. Energy. 2007, 32(5):795-802.
[3]Li, W.W.; Yi, P.T.; Zhang, D. Sustainability evaluation of cities in northeastern China using dynamic TOPSIS-Entropy methods. Sustainability.2018, 10(12), 4542.
(2) We have added a description of sustainability in the Literature Review. The following revisions were made.
2. Literature Review
2.2. Policy targets
According to a review of the literature, few studies have specifically attempted to investigate the sustainable of policy in governments at all levels. Most of the studies have focused on performance evaluation analysis of talent policy, but performance evaluation is a comprehensive measurement of talent policy targets. The policy targets are the expected direction of policy implementation, which is the purpose of policy instruments [11]. Different policy instruments can play different roles in the stage of the sustainable of policy [12]. Therefore, the targets of policy instruments should be considered while analyzing the type of policy instruments. Sun (2014) divided the talent policy evaluation system into four aspects: talent quantity, talent quality, talent input, and talent output [13]; Serban and Andanut (2014) expounded the relevant indicators of the global talent competitiveness index, which are mainly divided into input indicators and output indicators [14].
2.3. Policy strength
The strength of the policy is determined by the administrative level of the policy issuing agency. Usually, the level of policy promulgating units determines the policy strength, and the higher the level of the policy promulgating units, the stronger the strength of the policy promulgating units. This phenomenon reflects that the higher the government attaches importance to the sustainable development of talent. The weaker policies are issued by the policy promulgating agencies at lower levels, which also reflects that the lower levels of government attach importance to the sustainable development of talent.
(3) In the results, we have added the analysis of sustainability from three perspectives: policy instruments, policy targets, and policy strength. The following revisions were made.
4. Results
4.2. Policy instruments
Government's dependence on various policy instruments is different in promoting sustainable development of talent. The government is inclined to continually optimize the external environment for talent development through indirect methods. The low frequency of demand-side policy instruments will lead to the absence of a stable market environment for the sustainable development of talent.
4.2.1 Overuse of environmental policy instruments for sustainable of policy
Although there are many measures, a vacuum zone remains, such as how to ensure a fair talent-sustainable development environment and carry out the follow-up work after the introduction of high-level talent. With the advancement of society, the quality of both sexes is constantly improving, and the development guarantee of female talent should also be a critical aspect of talent policy planning. Moreover, the increasingly prominent role of talent has made the competition among high-level talent intensify. Therefore, how to create a satisfactory policy environment for the sustainable development of talent while performing a satisfactory job of talent introduction is also a focus of follow-up talent work.
4.2.2 Insufficient use of supply-side policy instruments for sustainable of policy
The policy of talent cultivation in supply-side policy instruments shows a greater advantage. One reason is that in the era of the knowledge-based economy, higher requirements are put forward for the cultivation of talent. It is necessary to cultivate their higher learning and innovation abilities to make them achieve sustainable development. Talent cultivation in the new era is imperative and different types of talent require different training methods. The introduction of talent cultivation policy reflects that the government follows the growth of talent and attaches importance to the sustainable development of talent. A very direct and effective means to promote the sustainable development of talent is by cultivating talent in different fields in an appropriate manner. These factors show that the policies of talent infrastructure construction and public service system construction in Sichuan Province are still insufficient, and the public services available in the process of the sustainable development of talent are not perfect. In the information age, the support and availability of talent information is the key to the sustainable development of talent.
4.2.3 Less use of demand-side policy tools for sustainable of policy
Demand-side policy instruments have ensured sustainability in a certain manner and broadened the source of talent. In the sustainable development of talent, demand-side policy instruments are more direct and faster than environmental policy instruments; they improve the quality of talent and ensure the sustainable development of talent. However, demand-side policy instruments have the lowest proportion, and the specificity and clarity of the policy content are weak. The inadequate application of demand-side policy instruments reduces their instrumental value in creating stable markets for the sustainable development of talent.
4.3. Policy targets
This study analyzes the role of talent policy targets in promoting sustainable development of talent and searches for policy instruments that must be strengthened in talent policy targets.
Sichuan Province promotes the sustainable development of talent mainly through talent innovation and the improvement of talent size, which is closely related to the competition of talent in various provinces and cities. The impetus of innovation is the main line of the sustainable development of talent. By strengthening talent innovation and expanding the talent size, talent will become the foundation of reform and innovation, enhancing its sustainability. Talent efficiency and talent flow account for almost the same proportion of policy targets. These two policies are mainly reflected in the evaluation of talent value through talent efficiency and ultimately promote the sustainable development of talent. Generally, in promoting the sustainable development of talent, the policy targets mainly use environmental policy instruments.
4.4. Policy strength
In Figure 4, the average strength of the policies issued by Sichuan Province since 2013 has shown an increasing trend, which reflects the importance of the sustainable development of talent, and also shows that the Sichuan government has increased its awareness of the importance of the sustainable development of talent. However, the policy has experienced ups and downs, which reflects the absence a scientific and systematic understanding of the sustainable development of talent. The understanding of the law to promote the sustainable development of talent is not sufficiently deep; thus, the sustainability and coherence of policies must be strengthened.
4.5. Cross-analysis of policy instruments-policy targets-policy strength
Therefore, we observe an inconsistency between policy instruments and policy targets in the current policy system of promoting the sustainable development of talent in Sichuan Province, mainly because no long-term guidance in the policy system.
[Comment 2]: English language and style are fine/minor spell check required.
[Response]: We must apologize for the poor writing. We employed a native English speaker, who is a professional language editor from Elsevier Webshop Support (webshop_support@elsevier.com), to improve the linguistic quality. Please check the certification files at the end of the response letter.
[Comment 3]: The introduction must be improved by adding the sufficient background and all relevant reference.
[Response]: Thank you for this comment. First, we have added a description of the sustainable development of talent in the introduction. Second, we have added relevant literature [1], [2], [3]. The following revisions were made.
1. Introduction
This series of questions has triggered our thinking, that is, it is unsustainable to solve the shortage of talent through policy adjustment. The policy focuses on solving short-term problems, and talent is a hundred-year plan, which should be solved through systems and legislation. The preferential policies such as settlement, housing, and financial support are tempting, but in the long run, it is not sufficient to support the sustainable development of talent. For second-tier and third-tier cities, they need to introduce talent and improve institutional mechanisms. Designed according to its own conditions, it is the best talent for the local talent; otherwise, it is a waste of the resources invested and the talent itself, which leads to the unsustainable development of talent.
An effective evaluation of sustainability is critical because it provides a useful framework for improved decision-making on all undertakings, such as policies, plans and programs, and physical undertakings [1]. However, sustainability is not easily measurable because it is not directly indicated as a natural or direct consequence of the reading of the indicators [2]. A widely accepted approach is about developing a framework or model by involving a number of sustainability criteria [3].
[1] Gibson, R.B.; Robert, B. Beyond the pillars: sustainability assessment as a framework for effective integration of social, economic and ecological considerations in significant decision-making. Journal of Environmental Assessment Policy and Management. 2006, 8(3), 259-280.
[2] Gagliardi, F.; Roscia, M.; Lazaroiu, G. Evaluation of sustainability of a city through fuzzy logic. Energy. 2007, 32(5), 795-802.
[3] Li, W.W.; Yi, P.T.; Zhang, D. Sustainability evaluation of cities in northeastern China using dynamic TOPSIS-Entropy methods. Sustainability.2018, 10(12), 4542.
[Comment 4]: The research design isn’t appropriate and the methods are not adequately described.
[Response]: Thank you for this comment. First, we have added the introduction of content analysis and added relevant literature [22], [25], [26], [29], [30]. Second, we have added a more detailed introduction to the data collection. The following revisions were made.
3. Methods
3.1. Research methods
Content analysis is a scientific, objective, and quantitative analysis method for different sample information content, and its concept was first proposed by Berelson [22]. The content analysis method combines qualitative and quantitative analysis of text content to clearly analyze essential facts and related trends of relevant topics within the policy documents [25], which has been used by many scholars to analyze policy. For example, Farchi and Salge (2017) used the content analysis method to trace the evolution of the innovation concept in policy discourse based on the analysis of 21 key policy documents published or commissioned by the English Department of Health between 1948 and 2015 [26]. Ledoux, et al. (2018) used content analysis to review the policy framework for migrants' access to healthcare in Spain, Portugal, and Ireland, countries with a long history of immigration, to identify lessons to be learned for policies on migrants' health [29]. Laestadius et al. (2019) examined e-liquid marketing on the visual social media platform Instagram, on which users have created significant amounts of e-cigarette-related content with the content analysis [30]. This paper also used content analysis to analyze whether talent policy can promote the sustainable development of talent.
3.2. Data collection
And (4) with the gradual development of the concept of the sustainable development and emphasis on talent, the number of talent policies has increased, as has the number of departments issuing such policies. A number of departments had jointly issued policy texts
[22] Berelson. B.B. Content analysis in communication research. American Political Science Association. 1952, 46(3), 869.
[25] Krippendorff, K. Content analysis: An introduction to Its methodology. Sage, Thousand Oaks, CA. 2012.
[26] Farchi, T.; Salge, T.O. Shaping innovation in health care: A content analysis of innovation policies in the English NHS, 1948–2015. Social Science & Medicine. 2017, 192,143-151.
[29] Ledoux, C.; Pilot, E.; Diaz, E.; Krafft, T. Migrants’ access to healthcare services within the European Union: a content analysis of policy documents in Ireland, Portugal and Spain. Globalization and Health. 2018, 14(1), 57.
[30] Laestadius, L.I.; Wahl, M.M.; Pallav, P.; Cho, Y.I. From apple to Werewolf: A content analysis of marketing for E-liquids on Instagram. Addictive Behaviors. 2018, 91, 119-127.
[Comment 5]: The results are not clearly presented.
[Response]: Thank you for this comment. We have added a description of the results to make them more intuitive and thematic. The following revisions were made.
4. Results
4.2. Policy instruments
Government's dependence on various policy instruments is different in promoting sustainable development of talent. The government is inclined to continually optimize the external environment for talent development through indirect methods. The low frequency of demand-side policy instruments will lead to the absence of a stable market environment for the sustainable development of talent.
4.2.1 Overuse of environmental policy instruments for sustainable of policy
Although there are many measures, a vacuum zone remains, such as how to ensure a fair talent-sustainable development environment and carry out the follow-up work after the introduction of high-level talent. With the advancement of society, the quality of both sexes is constantly improving, and the development guarantee of female talent should also be a critical aspect of talent policy planning. Moreover, the increasingly prominent role of talent has made the competition among high-level talent intensify. Therefore, how to create a satisfactory policy environment for the sustainable development of talent while performing a satisfactory job of talent introduction is also a focus of follow-up talent work.
4.2.2 Insufficient use of supply-side policy instruments for sustainable of policy
The policy of talent cultivation in supply-side policy instruments shows a greater advantage. One reason is that in the era of the knowledge-based economy, higher requirements are put forward for the cultivation of talent. It is necessary to cultivate their higher learning and innovation abilities to make them achieve sustainable development. Talent cultivation in the new era is imperative and different types of talent require different training methods. The introduction of talent cultivation policy reflects that the government follows the growth of talent and attaches importance to the sustainable development of talent. A very direct and effective means to promote the sustainable development of talent is by cultivating talent in different fields in an appropriate manner. These factors show that the policies of talent infrastructure construction and public service system construction in Sichuan Province are still insufficient, and the public services available in the process of the sustainable development of talent are not perfect. In the information age, the support and availability of talent information is the key to the sustainable development of talent.
4.2.3 Less use of demand-side policy tools for sustainable of policy
Demand-side policy instruments have ensured sustainability in a certain manner and broadened the source of talent. In the sustainable development of talent, demand-side policy instruments are more direct and faster than environmental policy instruments; they improve the quality of talent and ensure the sustainable development of talent. However, demand-side policy instruments have the lowest proportion, and the specificity and clarity of the policy content are weak. The inadequate application of demand-side policy instruments reduces their instrumental value in creating stable markets for the sustainable development of talent.
4.3. Policy targets
This study analyzes the role of talent policy targets in promoting sustainable development of talent and searches for policy instruments that must be strengthened in talent policy targets.
Sichuan Province promotes the sustainable development of talent mainly through talent innovation and the improvement of talent size, which is closely related to the competition of talent in various provinces and cities. The impetus of innovation is the main line of the sustainable development of talent. By strengthening talent innovation and expanding the talent size, talent will become the foundation of reform and innovation, enhancing its sustainability. Talent efficiency and talent flow account for almost the same proportion of policy targets. These two policies are mainly reflected in the evaluation of talent value through talent efficiency and ultimately promote the sustainable development of talent. Generally, in promoting the sustainable development of talent, the policy targets mainly use environmental policy instruments.
4.4. Policy strength
In Figure 4, the average strength of the policies issued by Sichuan Province since 2013 has shown an increasing trend, which reflects the importance of the sustainable development of talent, and also shows that the Sichuan government has increased its awareness of the importance of the sustainable development of talent. However, the policy has experienced ups and downs, which reflects the absence a scientific and systematic understanding of the sustainable development of talent. The understanding of the law to promote the sustainable development of talent is not sufficiently deep; thus, the sustainability and coherence of policies must be strengthened.
4.5. Cross-analysis of policy instruments-policy targets-policy strength
Therefore, we observe an inconsistency between policy instruments and policy targets in the current policy system of promoting the sustainable development of talent in Sichuan Province, mainly because no long-term guidance in the policy system.
[Comment 6]: The conclusions are not supported by the results.
[Response]: Thank you for this comment. According to the unsustainable problem of talent policy in the result, the improvement measures were put forward to promote the sustainable development of talent. The following revisions were made.
6. Conclusion and policy recommendations
This study constructs an analytical framework from the dimensions of policy instruments, policy targets, and policy strengths to examine policy sustainability. The results show that in the supply-side policy instruments, many "capital investment" policy measures have been used, but the role of "talent information support" in the role of talent development has been neglected. Among the environmental policy instruments, "strategic measures" are used more frequently, and the economic leverage of "tax finance" has not been fully used. Among the demand-side policy instruments, "talent introduction" and "trade control" are used more frequently, but the application of "overseas talent agencies" has not occurred. Policy targets focus on talent innovation and talent efficiency, but pay less attention to talent flow; Policy strengths are in line with strategic adjustments, but policies are not sufficiently sustainable.
Optimize the use of supply-side policy instruments and strengthen the supportive role of health care factors. The government should pay attention to the supporting role of public services, infrastructure construction, and other health factors for the sustainable of policy.
Promote the use of demand-side policy instruments and playing a direct role in the sustainable of policy. Compared with supply-side policy instruments and environmental policy instruments, demand-side policy instruments have a more direct role in the sustainable development of talent.

Reviewer 2 Report
I read with great interest the manuscript entitled “Content analysis of talent policy on promoting sustainable development of talent: Taking Sichuan province as an example”. The paper is fairly organized, while the exposition can be improved and the discussion should indicate the way ahead.
Some of my reservations are:
1) General comment, the manuscript will be improved by a professional editing, there are quite a few cases of inappropriate (scientific) style of writing, such as lines, 25-26, 39-40, 148-149, 199-200, 210-212
2) Only three journal papers only back up the methodology, namely {21], [23-24]. None of the seminal works have been cited, let alone the scopus lists some 6000 papers with the “content analysis” on their titles.
3) Conclusion and discussion hardly contains any discussion (in the sense of trying to explain the observed results).
Author Response
[Comment 1]: General comment, the manuscript will be improved by a professional editing, there are quite a few cases of inappropriate (scientific) style of writing, such as lines, 25-26, 39-40, 148-149, 199-200, 210-212. English language and style are fine/minor spell check required.
[Response]: We must apologize for the poor writing. We employed a native English speaker, who is a professional language editor from Elsevier Webshop Support (webshop_support@elsevier.com), to improve the linguistic quality. Please check the certification files at the end of the response letter.
[Comment 2]: Only three journal papers only back up the methodology, namely {21], [23-24]. None of the seminal works have been cited, let alone the scopus lists some 6000 papers with the “content analysis” on their titles.
[Response]: Thank you for pointing this out. We have updated the references [22], [25], [26], [29], [30]. The following revisions were made.
[22] Berelson, B.B. Content analysis in communication research. American Political Science Association. 1952, 46(3), 869.
[25] Krippendorff, K. Content analysis: An introduction to Its methodology. Sage, Thousand Oaks, CA. 2012.
[26] Farchi, T.; Salge, T.O. Shaping innovation in health care: A content analysis of innovation policies in the English NHS, 1948–2015. Social Science & Medicine. 2017, 192, 143-151.
[29 ] Ledoux, C.; Pilot, E.; Diaz, E.; Krafft, T. Migrants’ access to healthcare services within the European Union: a content analysis of policy documents in Ireland, Portugal and Spain. Globalization and Health. 2018, 14(1), 57.
[30] Laestadius, L.I.; Wahl, M.M.; Pallav, P.; Cho, Y.I. From apple to Werewolf: A content analysis of marketing for E-liquids on Instagram. Addictive Behaviors. 2018, 91, 119-127.
[Comment 3]: Conclusion and discussion hardly contains any discussion (in the sense of trying to explain the observed results).
[Response]: Thank you for this comment. We have added a discussion to the manuscript. The following revisions were made.
5. Discussion
5.1 Policy instruments and sustainable of policy
Our study shows that in the supply-side policy instruments, many "capital investment" policy measures have been used, but the role of "talent information support" in the role of talent development has been neglected. Among the environmental policy instruments, "strategic measures" are used more frequently, and the economic leverage of "tax finance" has not been fully used. Among the demand-side policy instruments, "talent introduction" and "trade control" are used more frequently, but the application of "overseas talent agencies" remains blank.
In the information age, the acquisition and effective use of talent information is the basis and guarantee of doing a good job in talent work. All types of talent databases and information databases are the support and support the effective docking of supply and demand information of scientific and technologically innovative talent, and it can also provide more complete services for the exchange and development of talent. To guarantee comprehensive information and perfect service, we should integrate the information of talent, projects, policies, funds, and other factors in the content construction of information platforms, use the systematic and interactive role of information platforms, and expand the amount of information. We should also provide channels for information exchange and interaction between the supply and demand sides. In terms of the type of information platform construction, it should include the information base serving the orderly flow of talent and the scientific allocation of talent as well as establish an integrated platform to promote the flow of information among industries, scientific research, and innovation work and facilitate the use and transformation of scientific and technological achievements. We also found that the imperfect and incomplete policy content may lead to the imperfection of talent information support. There are no specific requirements for the quality of information release and the perfection of platform construction in the policy items and no regulations and explanations for the comprehensiveness of information release content and the interactive role of platforms. The problem is that the content of information platform issuance is not comprehensive and perfect, and the quality of platform construction must be further improved.
In Table 3, we observe that the main purpose of environmental policy instruments is to provide a satisfactory policy environment for the sustainable development of talent and to open channels for the sustainable development of talent through indirect means. Compared with other policy instruments, the effectiveness of this policy instrument is slower, and indirectness is its main feature. Environmental policy instruments are indispensable, but there has been overuse of these instruments in Sichuan Province. Moreover, in environmental policy tools, policy tools are highlighted as strategic measures; on the one hand, this is caused by the short-term behavior of the government, and on the other hand, it is caused by the imperfection of the system. The government attaches great importance to the construction of a talent management system and mechanism, formulates comprehensive policies to guide the use and selection of talent, improves the degree of legalization, and effectively standardizes and optimizes the sustainable development of talent. However, "tax incentives" have not been a focus at the level of policy-making and has been applied less often, and the policy ambiguity is strong. As a critical guarantee regarding the sustainable development of talent work, the absence of "tax finance" will significantly reduce the effectiveness of environmental policy instruments. Therefore, policies in this field must be proposed and perfected.
To some extent, demand-side policy instruments are more direct, effective, and efficient than supply-side and environmental policy instruments in promoting the sustainable development of talent. From service outsourcing to trade control, the direction is clear, that is, it is the most direct means to promote the development of talent undertakings and ensure the sustainability of talent development. The establishment and management of overseas talent institutions have broadened the sources and channels of talent, guaranteed the quality of talent, and promoted the exchange and flow of talent. However, from the analysis results, the use of demand-side policy instruments is generally low, and the use of trade control, service outsourcing, overseas talent institutions, and other instruments is particularly inadequate. At present, Sichuan Province's talent management work is mostly in the stage of self-service and does not have the ability and experience to train and use social forces to engage in talent service work, and the policy guidance in this respect is not strong. Most of the policy documents that have been issued have only programmatic provisions, advocating cooperation between the government, enterprises, and social institutions to improve the quality and efficiency of personnel services and reduce the cost of social management through various means such as outsourcing and crowdsourcing. However, the specific means and methods of service outsourcing are not specified in detail, which leads to the inoperability of follow-up policies. The use of demand-side policy instruments will make industrial development and markets more closely integrated and embedded, to create a more favorable industrial and social environment for the development of talent research work and play a role in promoting the sustainable development of talent. By contrast, if the absence of demand-side policy instruments and general content weakens the overall guiding role of provincial policies; thus, the follow-up improvement of demand-side policy instruments should be the focus of government policy-making efforts.
5.2 Policy targets and sustainable of policy
The construction of an innovative talent team is of great importance. The macro level affects national economic development and international competitiveness, and the micro level relates to the development of various organizations. Therefore, the management of scientific and technologically innovative talent is a global endeavor, which requires strategic vision and overall planning. To determine the policy targets is to forecast the total supply and demand of scientific and technologically innovative talent, to analyze the supply and demand of various professional fields, to determine the source of scientific and technologically innovative talent, and to put forward the corresponding attracting measures, considering the needs of the current and long-term development of the region. The scientific clarity of targets is related to the implementation of future policies and the sustainable development of talent.
The study finds that the overall planning and assumptions for the flow of talent are less mentioned in the policy. It is also determined by the nature of the policy targets that generally belong to the strategic policy category. However, among the only policy targets, there are more ideas and plans for talent innovation and improvement of talent efficiency but fewer plans for talent flow and investment, and imperfect planning system and content. Policy targets are scattered in various policies. First, they weaken the importance of policy targets and are not conducive to the follow-up implementation of policies. Second, the policy is not systematic and has a certain impact on the scientific nature of the goal of talent-team building. In addition, problems remain in the policy, such as unclear goal setting, vague content and, difficulties with measurement.
5.3 Policy strength and sustainable of policy
We also observed that the policy strengths are in line with strategic adjustments, but policies are not sufficiently sustainable. The increase in policy intensity is significantly higher than the number of policies, which indicates that in these peak years, Sichuan Province introduced more talent policies and issued higher levels of policies. Additionally, most of the personnel policies with a strong legal effect have been published and implemented at an early stage. Weak legalization of policy is unfavorable to the stability and continuity of talent policy and inevitably leads to the convergence or conflict between talent policies because of the absence of guidance and norms of programmatic policies. Policy intensity that is too low is detrimental to the systematic and systematic distribution of talent policy. In addition, in 2015 and 2016, although more talent policies were issued, most were the opinions of the commissions, bureaus, and offices under the jurisdiction of the Sichuan Provincial Government, which led to the low average policy strength. This analysis shows that the change in the annual average effectiveness of talent policy in Sichuan Province is less affected by the legal effect of the policy. The poor policy strength will hinder the strategic and integral design of talent policy system, and cannot produce more effective impetus for the development of talent.

Reviewer 3 Report
I appreciate the beneficial complements made by the author, which has contributed to increasing the level of relevance, originality and analytical robustness of the paper.Author Response
[Comment 1]: English language and style are fine/minor spell check required.
[Response]: We must apologize for the poor writing. We employed a native English speaker, who is a professional language editor from Elsevier Webshop Support (webshop_support@elsevier.com), to improve the linguistic quality. Please check the certification files at the end of the response letter.
[Comment 2]: I appreciate the beneficial complements made by the author, which has contributed to increasing the level of relevance, originality and analytical robustness of the paper.
[Response]: Thank you for your comments. We appreciate your recognition of this manuscript.

Round 2
Reviewer 1 Report
I was desolate to write the previous review, I'm happy to write this. Congratulations. All suggestions are implemented.
This manuscript is a resubmission of an earlier submission. The following is a list of the peer review reports and author responses from that submission.
Round 1
Reviewer 1 Report
It would be advisable to present some examples of good international practices. The knowledge level reached in this thematic area needs to be more fully set up. We recommend a SWOT analysis of the state of play at this time. It is necessary to emphasize more clearly the factors on which the success of such policies depends.
Reviewer 2 Report
I read with great interest the manuscript entitled “Content analysis of talent policy on promoting sustainable development of talent:Taking Sichuan province as an example”. The paper is not well organized, the exposition can be improved and the discussion does not do justice to the analysis.
Some of my reservations are, as they appear:
1) General comment, the manuscript will be improved by a professional editing, there are many language blemishes and not appropriate scientific style of writing
2) The dominance of Chinese References do not do justice the paper indicating a lack of familiarity with the wider literature/.
3) A more advance methodology is generally expected.
4) It is always a good idea to check the aim and scopes of a Journal to check whether your manuscript fits well with it.
Reviewer 3 Report
The topic analysed by authors is relevant. I think that is also innovative. At the same time, I think that the sustainability is provided in this work but I suggest the authors another work published in MDPI: Social Sciences. It is more linear to the objectives of this journal.
However, the current structure is weak:
- The scientific context is totally absent. There are no justification regarding the novelty of the work. Literature must support this aspect
- Abstract is weak
- Section 2 is valid.
- materials and methods. What is your novelty? what are limits? what is the flexibility of the model? what are other model used in this context.
- results are interesting
I can only reject this manuscript.